# DNA-based identification of plants and the genomic nature of plant species differences
Wu Huang [1,2,3 ✉], De-Zhu Li [4,5,6], Alexandre Antonelli [3,7,8,9], Christine D. Bacon[7], Lian-Ming Gao [4,5,6], Catherine Kidner [1,10], R. Toby Pennington [1,11], Douglas E. Soltis [12,13], Pamela Soltis [12,13], Jeannine Cavender-Bares [14], Camille Christe[15], Kyle G. Dexter [1,16,17], Yanqian Ding[1,2], Mario Durán-Castillo[2], Mario Fernández-Mazuecos [18], Elliot M. Gardner[19], Morgan R. Gostel[20], Margarita Hernandez [21], Andrew L. Hipp [22], Johanna R. Jantzen[12,13], Jacob B. Landis [23], Xiu-Qun Liu[24], Andre A. Naranjo [25], James Nicholls [1,26], Han-Tao Qin[4,5], Jeffrey P. Rose[27], Nicolas Salamin [28], Rowan Schley[11], Philipp M. Schlüter [29], Jessica D. Stephens[30], Matthew A. Streisfeld[31], Natascha D. Wagner [32], Xiao-Quan Wang [33], Qiu-Yun Jenny Xiang [34], Alex D. Twyford [1,2 ✉] & Peter M. Hollingsworth [1 ✉]

Telling species apart using DNA sequence data plays a key role in understanding, monitoring, and managing biodiversity. However, plant species discrimination is often difficult due to the complex nature of plant species boundaries. To inform future strategies for DNA-based identification of plants using the nuclear genome and to gain fundamental insights into the genomic nature of differences between plant species, we conducted a large-scale analysis mining data from 151 studies. Of the 1713 multiple-sampled species evaluated, 1202 resolved as monophyletic (70.2%). We then assessed the density of species-specific SNPs (SSSNPs) in the DNA sequence data - of the 462 species from 27 genera assessed in detail, there was a median density of 193 SSSNPs per Mb and 412 species (89.2%) had at least one SSSNP. Randomly sub-sampling the SNP data showed an asymptote in species discrimination with around 3000 randomly selected SNPs. Finally, we undertook a resampling of 6 target-capture datasets and showed that 1-9 pre-selected loci provided equivalent levels of species discrimination compared to hundreds of nuclear loci. These findings provide an important quantitative assessment of the genomic nature of differences between plant species and provide foundations for the development of enhanced approaches for high-resolution DNA-based plant species discrimination.

Accelerating species delimitation and identification is a pressing challenge for biodiversity science and conservation[1,2], given the importance of species as a fundamental unit of biodiversity and the rapidly accelerating rate of biodiversity decline[3]. DNA barcoding, using the mitochondrial *cytochrome oxidase 1* (*CO1*) gene has proved remarkably successful in discriminating among animal species[4,5]. In plants, the standard DNA barcoding approach based on two or three plastid genes[6], i.e., *rbcL*[7], *matK*[8], *trnH-psbA*[9] and the internal transcribed spacers (ITS)[10–12] of nuclear ribosomal DNA has proved useful for many different applications[13]. This barcoding approach, however, often provides resolution to species-groups, rather than uniquely discriminating closely related species[13]. A number of attributes of plant species and the current plant barcodes all contribute to this imperfect resolution. Plant species hybridise frequently[14] and introgression and incomplete lineage sorting (ILS) of the barcode-region-containing plastid genomes and rDNA loci are well documented[15–18] and this undoubtedly contributes to DNA barcode sharing among species. Furthermore, rapid speciation mechanisms including polyploidy and/or breeding system transitions can lead to morphologically recognised taxa that do not show corresponding divergence in standard barcoding loci[13]. Additional intrinsic limitations of the standard DNA barcode regions include (a) the relatively slow mutation rate of plastid genomes which impacts the rate of generation of taxonomically informative species-specific substitutions[19] and (b) the

https://doi.org/10.1038/s42003-026-09858-7                                                                                    **Article**

significant asymmetry in seed:pollen dispersal in most plant species, where limited seed dispersal constrains the likelihood of substitutions in predominantly maternally inherited plastid DNA from spreading throughout a species range[13,20].

Efforts to extend the standard plant barcode to include a wider coverage of the plastid genome provides a modest increase in discriminatory power, but is fundamentally limited by the fact that plastid genomes are typically maternally inherited and often do not track species boundaries[17,21]. Thus, although there is a multitude of benefits and insights garnered from the use of standard plant barcodes, there is also an outstanding research challenge to develop improved scalable DNA-based methods that can more reliably distinguish plant species[16,22].

Addressing this challenge involves tackling two closely related issues. Firstly, at a practical level, there is a need to identify alternative DNA sequencing strategies that target genomic regions with increased discriminatory power, such as multiple loci from the nuclear genome[16]. Secondly, and at a more conceptual level, there is a need to better understand the genomic nature of the differences between plant species, such that the design of any improved DNA assay is informed by the nature of the problem. Outstanding issues here include understanding the relative proportions of taxonomically recognised plant species that resolve as monophyletic versus those whose evolutionary history and mode of speciation result in non-monophyletic relationships among related species[23,24], even with numerous nuclear markers[25]. Likewise, the frequency and distribution of species-specific nucleotide substitutions in the nuclear genome are unknown: are species-specific substitutions common throughout the genome, or is the predominant signal between species one of frequency differences across multiple loci, with fixed differences being restricted to a small number of loci under strong selection[26]. Refining DNA-based identification approaches using the nuclear genome will be inherently easier if species monophyly is the norm and taxon-specific substitutions are common.

In this study, we compile and analyse datasets that have generated large amounts of nuclear sequence data from multiple individuals of multiple congeneric species to provide a first synthetic assessment of the generalities of the genomic nature of the differences among plant species. These groups span a range of attributes, such as including woody and non-woody species, where differing rates of molecular evolution might affect the frequencies of species-specific substitutions, and genera of various sizes, including recent species radiations where previous molecular marker studies found limited genetic differences at the species-level.

First, we investigate the general nature of nuclear genomic differences between plant species, asking: (i) Whether most named plant species resolve as monophyletic as assumed in many standard tests of species discrimination, or whether non-monophyly is common due to a lack of resolution, polyphyly or paraphyly (as expected with recent species divergence and reticulation, or parapatric speciation especially where the progenitor species have large effective population sizes[25]). (ii) Whether most plant species are characterised by the presence of frequent species-specific Single Nucleotide Polymorphisms (SSSNPs) in the nuclear genome, or whether SSSNPs are infrequent in most cases. We then examine the implications of these findings for the development of practical approaches for plant species discrimination using DNA data by (iii) assessing the minimum number of nuclear loci that are required, on average, to provide maximal levels of species discrimination. Collectively, these investigations provide insights into the nature and patterns of genomic differences among plant species and will facilitate the future development of genomic barcoding approaches in plants.

## Results
### How often are named plant species monophyletic?
We compiled 151 studies corresponding to 134 plant genera in which multiple nuclear loci of 1713 species were sequenced (Supplementary Data 1), including 123 angiosperm, 6 gymnosperm, 3 fern, and 2 moss genera. Data for two genera, *Linanthus* and *Leptosiphon*, were newly generated for this study. We calculated the Monophyletic Ratio, defined as the

proportion of monophyletic species out of the total number of species with multiple sampled individuals. Overall, 70.2% of species resolved as monophyletic. Among the 151 studies, 37 (24.5%) had 100% of the recognised species resolve as monophyletic, while 77 (51.0%) had less than 75% (Fig. 1A), with 70% of species resolving as monophyletic when averaged across studies. The Pearson Correlation Coefficient (r) between study size (number of species sampled) and monophyletic ratio was very weak ($r = 0.057$). No significant difference in monophyletic ratio was observed between plant groups with different lifeforms ($p = 0.81$, woody vs. herbaceous, Supplementary Data 2, Fig. 1B) or between studies using different sequencing techniques such as target capture, genome skimming, transcriptome sequencing, and RAD/GBS (two-sided Wilcoxon rank-sum test, p-values all > 0.05, Supplementary Data 2, Fig. 1C)[27].

### Are species-specific SNPs the norm or the exception?
To understand the genetic basis of species differences, we characterised the abundance of species-specific SNPs (SSSNPs) in 27 datasets, for which the sequence alignment data are accessible, representing 20 different seed plant families (Supplementary Data 3).

Overall, 16 datasets (59.3%) had at least one SSSNP in all species studied (Supplementary Data 4, Fig. S4). The density of SSSNPs ranges from 0 to 27,263 per Mb, with a median density of 193 SSSNPs per Mb over all assessed species (Fig. 2). In species resolved as monophyletic, the density of SSSNPs typically ranges from 22 - 5623 per Mb (excluding outlier taxa in the top or bottom 5%). In contrast, most non-monophyletic species have SSSNPs at a density between 0 and 648 SSSNPs per Mb (excluding outlier taxa in the top or bottom 5%) and the density of SSSNPs is as expected significantly different between monophyletic species and non-monophyletic species (Fig. S1, Wilcoxon signed-rank p-value < 0.001). There is also notable variation in the results for species from different genera, with SSSNP densities ranging from 650–7693 SSSNPs per Mb in genera such as *Linaria* (Plantaginaceae) to 0 and 4 SSSNPs per Mb in genera that have experienced recent speciation such as *Ophrys* (Orchidaceae)[28,29] (Supplementary Data 4, Fig. 2).

To check whether the observed density of SSSNPs was due to genuine biological signal, as opposed to random shared mutations among samples arising due to the large number of sampled nucleotides, we randomised the taxon assignment of species within genera and conducted the same analysis. The randomised-labelled data resulted in a significantly smaller number of SSSNPs for all 27 datasets compared with the original assignment (Wilcoxon signed-rank test average $p < 0.01$, Supplementary Data 5) and in most cases these randomised samples had zero SSSNPs.

### How much data are needed to achieve maximal species discrimination success?
To test the minimal number of loci needed to tell named species apart, we randomly and repeatedly sub-sampled SNP loci in 23 of the datasets (selected on the basis that the full dataset resolved at least two species as monophyletic) (Supplementary Data 6). From randomly subsampling 10 SNP loci and progressively increasing the number of loci, we evaluated the point at which maximal species discrimination success (monophyly) was achieved (i.e., reaching the point where an equivalent number of monophyletic species was resolved as in the full dataset).

Sub-sampling of these datasets (Fig. 3) showed that the number of species discriminated based upon monophyly increases sharply from 100 to 500 randomly drawn SNPs. The proportion of species resolved as monophyletic then begins to plateau at around 500 to 1500 SNPs. Over all genera, at ~1500 SNPs c. 90% of the species that resolve as monophyletic in the full datasets are resolved as monophyletic (Fig. 3A). At the within-genus level, at ~3000 SNPs almost all genera exhibit an asymptote in their levels of species discrimination and 21/23 genera (91%) have >85% of their distinguishable species being discriminated with 3000 randomly selected SNPs (Fig. 3B, Supplementary Data 7). Genera with a small number of multiple-sampled species (such as *Mimulus*, *Taxus*, and *Tsuga*) hit the maximum species discrimination at an earlier stage, i.e., increasing the data volume beyond c.

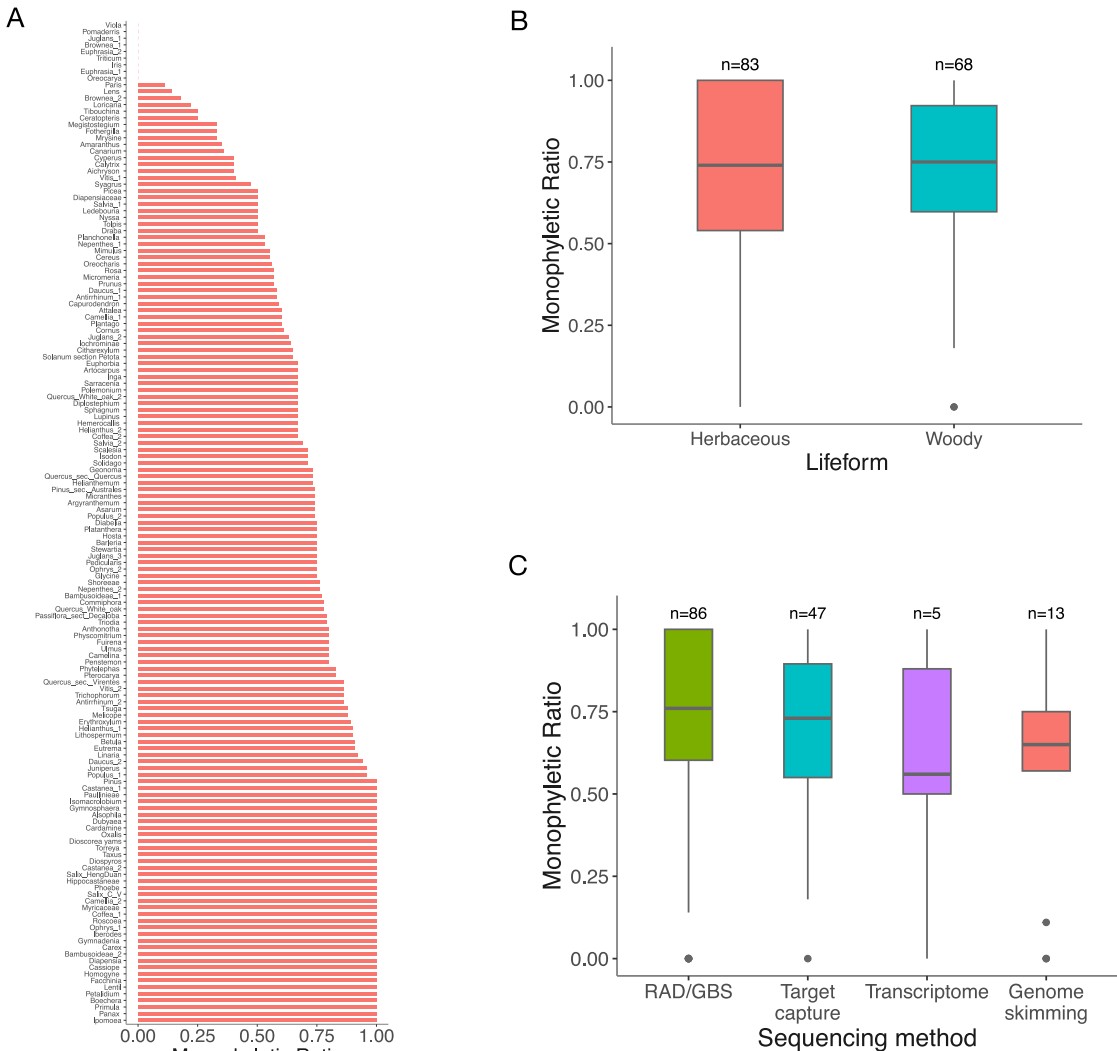

**Fig. 1 | Quantification of the proportion of plant species within a genus that resolve as monophyletic based on nuclear genomic data for 151 studies.**
**A** Proportion of species resolving as monophyletic in each study. **B** Comparison of plant groups with herbaceous and woody life forms. **C** Comparison of different sequencing methods. Boxplots show the median, lower, and upper quartiles, with whiskers extending to 1.5 times the interquartile range; the dots in (**C**) are the monophyletic ratio for each study.

3000 randomly selected SNPs does not increase the number of species resolved as monophyletic. Genera with more than 20 sampled species with multiple intra-specific samples, such as *Salix*, *Artocarpus*, *Geonoma*, and *Inga*, show a continued but slower increase from 500 to 1000 SNPs in the proportion of species resolving as monophyletic. The proportion of species resolving as monophyletic in some genera such as *Geonoma, Linaria*, and *Quercus* continued to slowly increase even after 7000 SNPs were sampled (Fig. S5).

### Do some genes show exceptional performance in species discrimination?

For six target capture datasets (Supplementary Data 8) where individual locus information was available, we performed sub-sampling at the level of individual genes rather than just the SNPs (this was not done on datasets with only small 'loci' such as RADseq). In five of the target capture datasets that had >400 genes sampled in their full dataset, four (*Inga, Tsuga, Polemonium*, and *Geonoma*) showed an asymptote in species discrimination at 100 genes or less, whereas in the fifth genus (*Artocarpus*) there was a slightly more protracted curve with 100 genes on average recovering 26 species as monophyletic and 200 genes recovering 28 species (Fig. S5).

To assess the performance of single loci in telling species apart, the frequency distribution of the number of species resolved by individual genes was plotted for the six genera with available datasets (Fig. S2). In five of these six genera, it approximates a normal distribution in the spread of performance of individual loci. In the remaining genus, *Polemonium*, most loci showed low levels of species discrimination with only a few loci being individually able to distinguish more than one species. In all datasets, there are clearly some loci that are much better than others in distinguishing species. For example, in *Geonoma*, the locus *LOC105045005* alone resolves 30 species as monophyletic (Supplementary Data 9), which is more than the number of species distinguished by using all 795 loci (which resolved 28 species as monophyletic). In four of the genera, the best single locus gave equivalent resolving power as the full dataset (Fig. S2; Table 1). Only in *Inga* and *Capurodendron* is there a bigger discrepancy between the efficacy of the best-performing locus and the total dataset. In *Inga*, the best-performing locus distinguished 31 species compared to 45 species from the full dataset; in *Capurodendron*, the best-performing locus distinguished 13 species, compared to 20 species with the full dataset. But even in the case of these two genera, the nine and seven most informative genes resolved as many species as monophyletic as 810 genes for *Inga* and 615 genes for *Capurodendron* (Table 1).

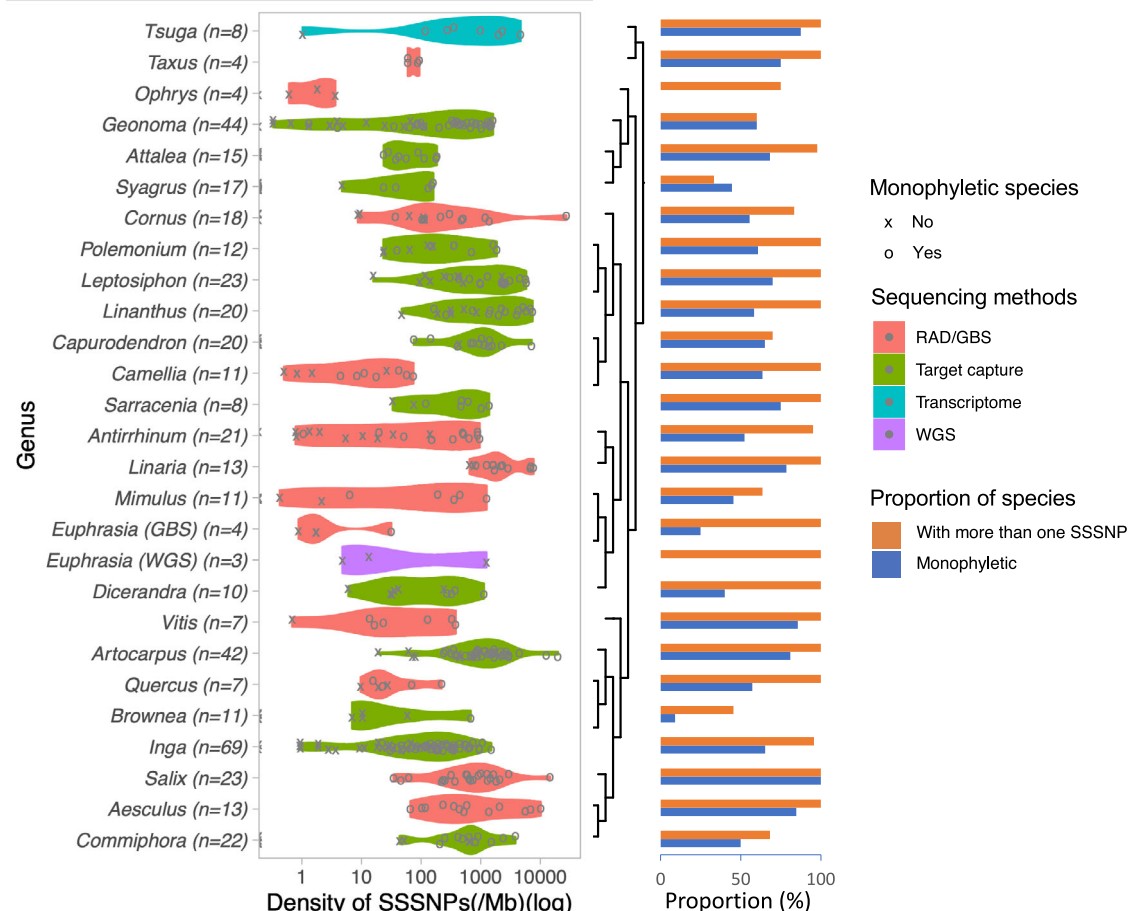

**Fig. 2 | Distribution of the density of species-specific SNPs (SSSNPs) for 27 datasets from 26 genera (left panel, log-transformed), with the phylogenetic relationships among study taxa (middle panel), and the proportion of species** that have more than one SSSNP (right panel). The right panel is the proportion of species in each genus that have more than one SSSNP (orange bars) and the proportion of species that resolve as monophyletic in that genus (blue bars).

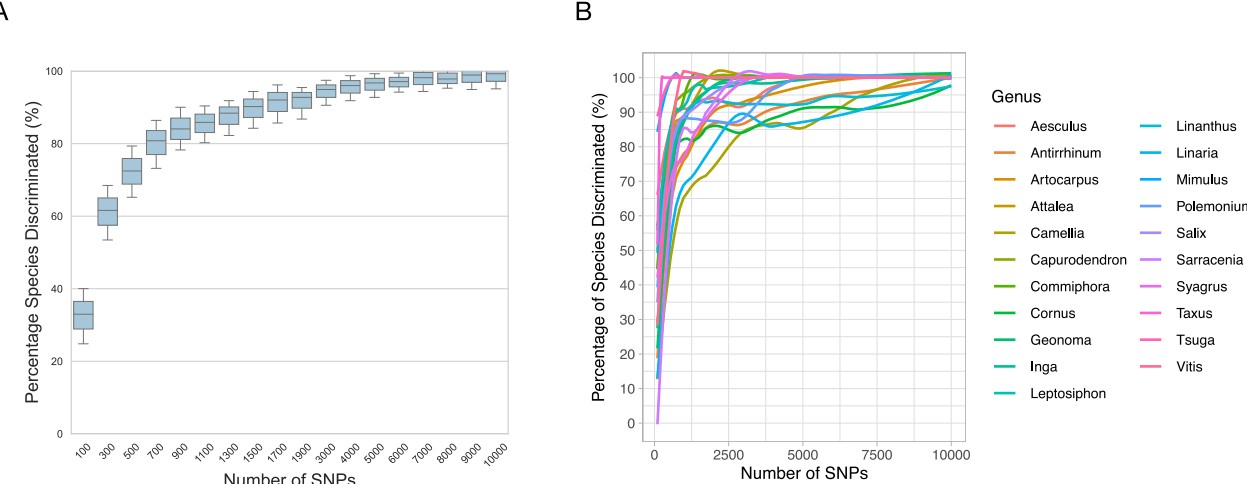

**Fig. 3 | The proportion of species discriminated using different numbers of sub-sampled SNPs. A** Percentage of species discriminated across 23 datasets (n = 23). The y-axis shows the percentage of species discriminated across all genera as a proportion of the total number of species resolved in the full dataset based upon monophyly. The x-axis is the number of SNPs randomly sub-sampled with 50 replicates for each sub-sampling interval. The intervals are increments of 200 SNPs between 100 to 1900 SNPs and 1000 SNPs between 3000 to 10,000 SNPs. **B** The proportion of species discriminated with increasing numbers of randomly drawn SNPs for each genus separately.

**Table. 1 | The minimal number of best-performing loci required to match the resolution of the full dataset, as measured by the number of species resolving as monophyletic**

| Genus | Number of loci in the dataset | Number of species resolved as monophyletic with all loci/Total number of multiple-sampled species | Minimum numbers of loci to achieve maximum species resolution | Number of species resolved as monophyletic by minimum number of loci |
|---|---|---|---|---|
| *Artocarpus* | 517 | 34 / 42 | 1 | 27 |
| *Capurodendron* | 615 | 13 / 20 | 7 | 14 |
| *Geonoma* | 795 | 30 / 44 | 1 | 30 |
| *Inga* | 810 | 45 / 69 | 9 | 44 |
| *Polemonium* | 360 | 7 / 12 | 1 | 7 |
| *Tsuga* | 881 | 7 / 8 | 1 | 7 |

We then examined whether there was a correlation between loci that were maximally informative in terms of discriminatory power (i.e., efficacy in recovering monophyly, density of SSSNPs) versus levels of nucleotide diversity (Supplementary Data 10, Fig. S3). Of the five genera tested (*Capurodendron* was excluded due to variation in the number of individuals per locus), there was a variable and only partially significant relationship between nucleotide diversity of genes within a genus and the density of SSSNPs in those genes (average correlation coefficient $r = 0.269$; this was statistically significant in only 3/5 tested datasets: *Geonoma* $r = 0.154$, *Inga* $r = 0.493$, *Tsuga* $r = 0.529$; two-sided Wilcoxon rank-sum test, $p < 0.001$). In terms of the relationship between nucleotide diversity and the number of species resolving as monophyletic, this was again a weak and only partially significant relationship; the average correlation coefficient $(r) = 0.193$ and was statistically significant in three of the five datasets (namely *Geonoma* $r = 0.159$, *Inga* $r = 0.423$, *Tsuga* $r = 0.185$; two-sided Wilcoxon rank-sum test, $p < 0.001$). As expected, there was a stronger relationship between the density of SSSNPs and species monophyly for individual genes (average correlation coefficient $r = 0.413$ and statistically significant in all five genera ($p < 0.001$, two-sided Wilcoxon rank-sum test).

## Discussion

This study provides a synthetic quantification of the species discrimination signal in plant multi-locus nuclear DNA sequence datasets. It also provides foundational information for the future development of nuclear DNA barcoding approaches in plants.

The first question to address is what proportion of plant species resolve as monophyletic based on multi-locus nuclear sequence data? The concept of species-level monophyly is central to many methods of species discrimination (e.g., molecular operational taxonomic units—MOTUs[30] or barcode index numbers - BINS[31]) and is indeed central to certain species concepts (e.g., the phylogenetic species concept per Mishler and Donoghue[32]; referred to as the monophyletic species concept by Judd et al.[33]). In plants, various authors have identified high levels of non-monophyly from standard plastid and ribosomal barcoding regions[34,35] but it is difficult to know how much of this relates to the attributes of barcoding loci, versus a more general point about the species-level monophyly of plant species. In our synthesis of 1713 named species from 151 studies, we found 1202 species resolving as monophyletic (70.2%) based on nuclear genomic data. At the level of individual genera, only 37 of the 151 studies (24.5%) had all multiple-sampled species resolved as monophyletic. The theoretical expectations that a substantial portion of plant species will not resolve as monophyletic are well established[23,36]. Key factors likely to explain such non-monophyly are: (a) recent speciation or long generation times, or both, leading to a predicted long time to monophyly and the retention of ancestral polymorphisms (e.g., incomplete lineage sorting)[37], (b) founder speciation/ peripheral isolate speciation[38], (c) hybridisation and homoploid hybrid speciation[39], (d) polyploid speciation[40].

A wider point, and one not assessed in the current paper, is the degree to which non-monophyly is attributable to imperfect taxonomy. A study of 41,583 museum specimens of European Lepidoptera estimated ~23% of species to be non-monophyletic for the *CO1* barcode, with 57% of non-monophyly at the species level attributed to taxonomic issues including under-splitting, over-splitting, or other identification difficulties[41]. In plants, a recent study of *Capurodendron* shows that some polyphyletic species were in fact different species supported by good morphological characters with an added identification key[42]. Taxonomic issues will impact this study, with different taxonomic concepts applied to different groups, particularly in cases where species boundaries are controversial (e.g., in *Euphrasia*[43] and *Antirrhinum*[44]).

Notwithstanding the unquantified component of taxonomic error in this study, the estimate of ~30% non-monophyly quantified here is consistent with earlier hypotheses of the expected occurrence of monophyly based on morphological and/or geographical data. Crisp & Chandler[45] undertook a partial survey of two angiosperm families (Fabaceae, Proteaceae) and reported c 20% of plant species resolved as paraphyletic. Likewise, in a more general perspective, Rieseberg & Brouillet[23] hypothesised that about 50% of plant species are "products of geographically local speciation, and that close to one half of these are likely not to be monophyletic".

The estimate of 30% non-monophyly of named plant species recovered in the current study may change with additional sampling, and increased sampling of individuals within species, and species within genera. This has the potential for further disrupting patterns of monophyly and decreasing the proportion of species resolving as monophyletic. On the other hand, further sampling of monotypic or species-poor genera and/or genera with highly divergent species, would lead to higher levels of species monophyly, especially if such genera are currently under-represented in the literature due to being of lower intrinsic interest to systematists. Another factor which may ultimately reduce the overall proportion of plant species resolving as non-monophyletic is the possibility that extensive nuclear sequence datasets will identify cases where a substantial proportion of plant species non-monophyly is due to imperfect taxonomy, and that subsequent taxonomic clarification and taxonomic revisions will act in turn to increase the proportion of plant species names that are associated with monophyletic genetic lineages. Finally, there is likely to be a technical/analytical component to the proportion of species resolving as non-monophyletic. For multiple-sampled species, adding more sequence data or conducting detailed analyses for each study could improve resolution, potentially showing some species currently classified as non-monophyletic to be monophyletic.

Quantifying the frequency distribution of species-specific SNPs provides baseline information on the genomic nature of interspecific differences in plants and practical information regarding the ease of designing species-specific diagnostic assays.

In this study, the majority (89%) of tested species had at least one SSSNP (412/462 species from 27 datasets) even in complex groups such as *Salix* (willows, Fig. 2). This was true even for many species that resolved as non-monophyletic and 116 non-monophyletic species had at least some SSSNPs. In these cases, these SSSNPs may simply reflect plesiomorphies in the ancestral

species, and/or may be linked to regions of the genome under selection and thus linked to the cohesiveness of a species[43].

The cases where we did not detect SSSNPs may be caused by imperfect taxonomy as outlined above or reflect the genuinely complex nature of species boundaries themselves[46,47]. In particular, recently formed species may not show fixed differences throughout much of the genome due to homogenising gene flow from relatives or maintenance of ancestral polymorphism, with species-specific differences limited to a few regions of the genome underlying divergence. Such clustered SNPs underlying species differences are unlikely to be represented in most RAD or target capture datasets and are only likely to be identified with whole-genome sequencing[48].

In terms of the density of SSSNPs, there is clearly a wide distribution in the frequency of SSSNPs (Fig. 2), although the median value of 193 SSSNP per Mb (IQR 20–834) is a useful summary catching the fact that SSSNPs are not so common they are routinely found every few 100 bp, nor are they so rare that they require megabases of sequence data to detect. This distribution of SSSNPs is of interest, as a first approximation of what proportion of the genome shows fixed differences between species and how this relates to wider patterns of variation. It is of course subject to sampling density, sequencing approach and the species divergence patterns among the 27 datasets. We stress that the SSSNP density distribution presented here should be interpreted as a very broad-brush first approximation rather than a robust point estimate, and it will undoubtedly be refined as more whole genome datasets become available across a wide range of densely sampled plant groups.

There is a consistent and growing demand for effective species identification, from elucidating the species composition of mixtures such as in food authentication, pollen metabarcoding and diet analyses, to species-level diagnostics and identification in forensic cases and species monitoring programmes[49]. In all these examples, there is a clear benefit to enhancing levels of species discrimination above and beyond that achievable with the current standard plant barcodes.

The rapid pace of development in sequencing technologies and the associated flood of data from the nuclear genome provide great potential for improvements in DNA-based species identification and gives optimism for the future design and definition of new nuclear barcoding approaches for plants. A comparison of available datasets which compare multi-locus nuclear sequencing approaches with standard DNA barcodes (Supplementary Data 12) shows clear improvement in species discrimination in 10 out of 12 cases (with the remaining two cases showing no difference). In terms of designing optimal future standard methodologies for exploiting the nuclear genome, when there is wide availability of reference genomes and associated whole-genome resequencing data for all species, it may ultimately be possible to identify species simply by genome resequencing[50]. However, until that future is realised there is a premium on more selective approaches. The current study indicates that without a priori selection, ~200 loci/3000 SNPs represents the general point from which increases in species discrimination tails off. It is noteworthy that this asymptote is obtained relatively steeply as opposed to a slow progressive gain of more species being discriminated when more data are added. However, this amount of sequence data is clearly substantially higher than any of the current barcoding approaches and thus currently challenging to deploy at scale to very large sample sets. One promising non-targeted approach is kmer-based methods of species discrimination based on genome skimming, and recent studies show clear promise for exploiting shallow-pass genome skimming data from the nuclear genome of plants[50–52].

An alternative perspective is to focus on more targeted approaches. The current study shows that, for a specific group, 1 to 10 best-performing loci can be equally effective as the full datasets from which they were derived. This result is encouraging and there is a high premium on follow up studies to assess whether there are some regions of the genome that are consistently informative across taxonomic groups and will allow the maximal discrimination of species with a much smaller pool of loci. An obvious starting point for this approach is to explore whether any of the loci in the Angiosperms353 target capture set[53], which were developed for (and initially applied to) deeper-level phylogenetic studies, have sufficient resolving power at the species level. This targeted approach using a small number of loci is a promising route for development, though any assay with a modest number of loci will be unlikely to provide complete species resolution in complex groups characterised by contrasting modes and tempos of divergence. And regardless of the number of loci used, some situations will remain challenging, such as distinguishing recently formed autoploid species from their diploid progenitors[54].

## Concluding remarks

The standard plant DNA barcodes remain a powerful and widely used method for understanding and characterising plant species diversity. To increase levels of resolution by exploiting signal in the nuclear genome of plants, there is a pressing need for community collaboration to address outstanding data and infrastructure needs. Immediate priorities include focused densely sampled genome resequencing studies aimed at understanding the genomic nature of plant species differences, and optimisation of pipelines and analytical methods for routinely and robustly quantifying the degree to which multi-locus nuclear sequence data can tell plant species apart in the most cost-efficient fashion. Moving beyond the currently available barcodes will bring considerable benefits in revealing the proportion of named plant species that correspond to coherent genetic groupings and also offer enhanced resolution in environmental and ecological biomonitoring applications. This latter point is of pressing global importance given the ambitious targets articulated in the Kunming Montreal Global Biodiversity Framework.

## Materials and methods

Our workflow involves the following key steps

1) Compile datasets which sample multiple individuals from multiple congeneric species for multiple nuclear markers
2) Assess % species monophyly based on the original published analyses
3) For a subset of these datasets evaluate the density and distribution of species-specific SNPs (SSSNPs)
4) For a subset of these datasets randomly and repeatedly subsample the data and execute a rapid tree building process to see how many randomly selected SNPs/loci are required to recover the same levels of monophyly as the full dataset
5) For target capture datasets assess whether some genes show higher species discriminatory power than others and assess whether discriminatory power is correlated with nucleotide diversity.

## Compiling datasets

We compiled studies to assess the extent of plant species monophyly, with the criteria that they were published after 2013, sequenced three or more unlinked nuclear loci, included at least three individuals from multiple congeneric species and had a phylogenetic tree where species monophyly could be inferred. The average number of multiple-sampled species per study was 12 (range from 2–53). A total of 151 plant groups from the published literature or from collaborators were included (Supplementary Data 1). Studies were categorised by sequencing techniques, namely (1) Restriction site-Associated DNA sequencing (RAD-seq)[55] and its derivatives, (e.g., GBS[56], ddRAD-seq[57], 2b-RAD[58]); (2) Target Capture[59]; (3) Genome skimming[60]; (4) Transcriptome or exon sequencing[61,62]. For full criteria for inclusion see this protocol https://doi.org/10.17504/protocols.io.kxygx3z9og8j/v1[63].

Among these studies, 27 datasets were selected to further assess genomic differences between plant species (Supplementary Data 3). They were chosen because they have relevant metadata that link sequences of individuals to their species identities, a sequence alignment file in .fasta, .phylip, or .nex format or SNP matrix in .vcf or .fasta format, and a phylogenetic tree.

## Assessing general patterns of species monophyly

In this study, we focused on assessing the proportion of species where there is a clear phylogenetic signal of species-level monophyly, e.g., where all individuals within a species group together as a monophyletic unit in a phylogenetic tree. We evaluated 151 published datasets and manually recorded the number of species represented by more than one sampled individual that resolved as monophyletic, as a proportion of the total number of species in the dataset with more than one sampled individual. This assessment of monophyly was based on the phylogenetic trees presented in the original publications which represented the original authors' best estimates of phylogenetic relationships.

## Assessing the frequency distribution of species-specific Single Nucleotide Polymorphisms (SSSNPs)

To estimate the density and abundance of SSSNPs in the 27 datasets with relevant sequence files, we calculated the number (and density per Mb) of SSSNPs from the total dataset with a SSSNP defined as a SNP that was fixed in one species and different from all other congeners. More details on these analyses are provided in this protocol https://doi.org/10.17504/protocols.io.5qpvo33rzv4o/v1[64].

To group related genera within Fig. 2, a phylogenetic tree was estimated based on version 3.0 of the Tree Of Life explorer https://treeoflife.kew.org/tree-of-life.

## Assessing the minimum number of nuclear loci that are required to provide maximal levels of species discrimination

To evaluate the minimum number of random SNPs or loci required for species discrimination, subsampling of SNPs or loci was performed on the 23 genera where at least three species resolved as monophyletic (Supplementary Data 6). For each dataset, a UPGMA tree was generated for 50 random subsets of 200 SNPs and the proportion of species resolving as monophyletic recorded with Monophy[65]. This process was repeated by incrementally adding more data, with 200 additional SNPs until 2000 SNPs were obtained, followed by adding 1000 SNPs at each step, until an asymptote in species discrimination was reached. Datasets where multiple SNPs could be recovered as loci (e.g., some target capture datasets) used an initial dataset of 10 loci, with a step size of 10 loci until 100 loci were reached, then 40 loci until 300 loci were reached, and subsequently 100 additional loci at each further step. We selected UPGMA as the tree building method based on its speed, and our basic requirement for estimating species monophyly from large numbers of trees from the simulations across multiple datasets with no requirement for estimating branch lengths or resolving the deeper nodes in the trees.

Additional analyses were performed to assess whether a small number of *selected* loci could result in species discrimination (measured by species monophyly) that was equivalent to a larger *random* selection of loci. These analyses were performed on the six datasets with defined loci such as target capture and which had little missing data, meaning that RAD data were excluded. These datasets include target capture data for *Artocarpus*, *Capurodendron*, *Geonoma*, *Inga* and *Polemonium*, and transcriptome sequencing for *Tsuga* (Supplementary Data 8). Specifically, we assessed which individual genes recovered the maximum number of species as monophyletic, and which minimal combinations of individual genes could recover the same number of monophyletic species as the complete dataset. Further details are provided in sections 6 and 7 of our analysis protocol, available at: https://doi.org/10.17504/protocols.io.5qpvo33rzv4o/v1.

## Statistics and reproducibility

All statistical analyses were conducted to quantify patterns of species discrimination and genomic differentiation using multi-locus nuclear DNA datasets compiled from published studies and collaborator-provided data. Statistical tests were applied where appropriate to assess differences between groups or to evaluate whether observed patterns differed from random expectations.

Comparisons of monophyletic ratios among studies, life forms (woody vs. herbaceous), and sequencing approaches were conducted using two-sided Wilcoxon rank-sum tests. Correlations between study attributes (e.g., number of species sampled) and monophyletic ratios were assessed using Pearson correlation coefficients. Differences in the density of species-specific single nucleotide polymorphisms (SSSNPs) between monophyletic and non-monophyletic species were evaluated using two-sided Wilcoxon signed-rank tests. To assess whether observed SSSNP densities exceeded expectations under random species assignment, species labels were permuted within genera and SSSNP densities recalculated; observed and randomised values were compared using paired Wilcoxon signed-rank tests.

For analyses involving data resampling, reproducibility was assessed through repeated random subsampling. SNP- and locus-level subsampling analyses were performed with 50 independent replicates at each subsampling interval. For each replicate, species discrimination success was quantified as the number of species resolving as monophyletic, using UPGMA trees and the MonoPhy R package. Reported values represent the distribution across replicates, with medians and interquartile ranges shown where appropriate.

Replicates were defined as independent biological samples (individuals) within species, as provided in the original published datasets. Only species represented by two or more individuals were included in analyses of species monophyly and SSSNPs. Sample sizes, therefore, varied among studies and genera, reflecting the original experimental designs, with an average of 12 multi-sampled species per study (range 2–53). No technical replicates were generated as part of this study; all analyses were conducted on biological replicates or on resampled subsets of existing genomic data.

To assess the relationship between genes that showed high levels of discriminatory power, the density of SSSNPs and levels of nucleotide divergence, we calculated the correlation coefficient using CORREL function in excel and tested the significance via a Wilcoxon rank-sum test. The nucleotide divergence of each locus was calculate using nuc.div function in R package pegas[66].

## Sequence data generation and alignment for *Linanthus* and *Leptosiphon*

The majority of data for this study was drawn from existing published studies. However, two datasets are presented here for the first time. The data were generated using the following method in studies led by Jacob B. Landis. Leaf material for 172 taxa of *Leptosiphon* and *Linanthus*, as well as taxa from *Gilia* and *Phlox* that serve as outgroup accessions, were collected from both field collection and herbarium specimens at the following herbaria: California Academy of Sciences (CAS), Jepson Herbarium – UC Berkeley (JEPS), Rancho Santa Ana Botanic Garden (RSA), University Herbarium – UC Berkeley (UC), and University of Cal'ide (UCR). DNA was extracted from all samples following a modified CTAB extraction protocol[67].

Resuspended DNA from fresh material was then sonicated to a targeted length of 300 bp using a Covaris S220 sonicator (Covaris Inc., Woburn, MA, USA) following the manufacturer's suggested protocol. Resuspended DNA from herbarium accessions showed sufficient fragmentation so that sonication was not necessary. A total of 3–5 µg of DNA from each sample was sent to RapidGenomics (Gainesville, FL, USA) for Illumina library preparation with dual indexed barcodes. Targeted exon capture was conducted using MYbaits probes (MYcroarray, Ann Arbour, MI, USA) described by Landis et al.[68]. Capture products were pooled and distributed across three individual Illumina runs: 10 samples in a HiSeq 2000 (2 × 100 bp), 22 samples in a MiSeq 2 × 150 bp, and 148 samples in a NextSeq 2 × 150 mid-throughput run.

Raw reads were processed using the custom scripts described by Landis et al. (2016). Briefly, these scripts trim and filter the reads using cutadapt[69]. To pass the filtering parameters, sequences must have a minimum score of 20 and minimum length of 20 bp. Cleaned reads were then used in a BLAT[70] analysis to isolate plastome reads and on-target nuclear reads. For the 100 nuclear genes that were targeted, we obtained an average

of 14,048 on-target sequences for 172 taxa, ranging from 6 to 106,937 paired-end reads per taxon. In addition to the targeted nuclear genes, we obtained on average 31,154 reads matching the plastome, with a range of 48 to 249,734 paired-end reads per taxon.

Nuclear and plastid coding genes were then assembled to individual genes using default parameters of HybPiper[71]. For both nuclear and plastome data, genes were separated by using Burrows-Wheeler Aligner (BWA)[72] analysis against a reference file. Each gene was then aligned using MAFFT[73] (version 7.245) installed on the University of Florida Research Computing cluster using pairwise comparisons with 1000 iterations. Aligned sequences were then concatenated using SequenceMatrix[74]. Each concatenated sequence was then analyzed using PartitionFinder[75] (version 2.0) using a greedy algorithm and RAxML[76] (version 8) to find the best partition scheme for RAxML analyses. Raw reads for each accession were deposited in GenBank's Short Read Archive (http://www.ncbi.nlm.nih.gov/sra) under the NCBI Bioproject number PRJNA322057.

Further analyses consisted of separate matrices for both *Leptosiphon* and *Linanthus* of the following data sets: (1) concatenated matrix of 61 nuclear loci for both genera (referred to hereafter as the nuclear total evidence approach), (2) concatenated matrix of 22 nuclear loci for *Leptosiphon* and a concatenated matrix of 14 nuclear loci for *Linanthus* that each produced a well-resolved phylogeny in preliminary analyses (hereafter referred to as the reduced nuclear data set), and (3) concatenated matrix of 80 plastid protein-coding regions for each genus. All analyses were conducted using RAxML on the University of Florida Research Computing cluster using the partitions identified by PartitionFinder and a GTR + G substitution model. Support values were determined by running 1000 bootstrap replicates. The two species of *Gilia*, *G. brecciarum* subsp. *brecciarum* and *G. nevinii*, were used as outgroups.

To incorporate phylogenetic uncertainty in the character evolution analyses, separate Bayesian runs were conducted using MrBayes[77] (version 3.2.6) using the nuclear reduced data sets for each genus with the identified partitions from above. The *Linanthus* analysis was conducted for 5 million generations, while the *Leptosiphon* analysis was conducted for 7 million generations, each sampling every 1000 generations. The final 1000 trees of the posterior distribution were used as a sample of the Bayesian posterior distribution.

### Inclusion and ethics statement

This research was conducted in accordance with all relevant institutional, national, and international guidelines and regulations. No human participants, identifiable human data, or animal experiments were involved. All biological materials were obtained under appropriate collection permits and material transfer agreements, and their use complies with the Convention on Biological Diversity and the Nagoya Protocol on Access and Benefit-Sharing. We are committed to promoting transparency, open scientific exchange, and equitable collaboration. Local partners and institutions involved in sample access and knowledge exchange are acknowledged as research contributors and co-authors where appropriate. The project did not exclude any individuals or groups from participation on the basis of personal characteristics, and efforts were made to ensure accessible communication of the research findings.

### Data availability

The list of datasets used in this study can be found in supplementary Supplementary Data 13 with reference to publications listed in supplementary Supplementary Data 11. Sequence data, alignments, phylogenetic trees, and metadata are archived and publicly available in Zenodo https://zenodo.org/records/17603347 (https://doi.org/10.5281/zenodo.17603347). The repositories for raw sequences of the published studies are specified in the cited publications, with the exception of *Linanthus* and *Leptosiphon*, *Attalea* and *Syagrus*, of which raw reads for each accession were deposited in GenBank's Short Read Archive (http://www.ncbi.nlm.nih.gov/sra) under the NCBI Bioproject number PRJNA322057 and PRJNA1074667, respectively.

### Code availability

All scripts for computational analyses used in this study is released on GitHub and archived on Zenodo. The archived release is available at https://doi.org/10.5281/zenodo.18034794.

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

## Acknowledgements

This work was supported by a Darwin Trust of Edinburgh PhD studentship to Wu Huang; and also part funded by Horizon Europe under the Biodiversity, Circular Economy and Environment (REA.B.3); co-funded by the Swiss State Secretariat for Education, Research and Innovation (SERI) under contract number 22.00173, and by the UK Research and Innovation (UKRI) under the Department for Business, Energy and Industrial Strategy's Horizon Europe Guarantee Scheme, supported by Wellcome through a Darwin Tree of Life Discretionary Award (218328). The Royal Botanic Garden Edinburgh is supported by the Scottish Government's Rural and Environment Science and Analytical Services Division. The large-scale analysis was conducted on the UK crop diversity bioinformatics HPC facility (https://www.cropdiversity.ac.uk). For a full list of data resources and providers please see Supplementary Data 11.

## Author contributions

W.H., P.M.H., and A.D.T. designed research; W.H. designed and developed bioinformatic pipeline and performed data collection and analysis; D.Z.L., A.A., C.D.B, L.M.G., C.K., R.T.P., D.E.S., P.S., J.C.B., C.C., K.G.D., Y.Q.D., M.D.C., M.F.M., E.M.G., M.R.G., M.H., A.L.H., J.R.J., J.B.L., X.Q.L., A.A.N., J.N., H.T.Q., J.P.R., N.S., R.S, P.M.S., J.D.S., M.A.S., N.D.W., X.Q.W., Q.Y.J.W. contributed sequence data; and W.H., P.M.H., and A.D.T. wrote the paper with input from D.Z.L, L.M.G., P.S, D.E.S, R.T.P, and A.A. All authors reviewed the final version of the manuscript.

## Competing interests

The authors declare no competing interests.

## Additional information

[1]Royal Botanic Garden Edinburgh, Edinburgh, UK. [2]Institute of Ecology and Evolution, School of Biological Sciences, University of Edinburgh, Edinburgh, UK. [3]Royal Botanic Gardens, Kew, Surrey, UK. [4]Germplasm Bank of Wild Species, Kunming Institute of Botany, Chinese Academy of Sciences, Kunming, China. [5]Lijiang Forest Diversity National Observation and Research Station, Kunming Institute of Botany, Chinese Academy of Sciences, Lijiang, China. [6]Center for Interdisciplinary Biodiversity Research & College of Forestry, Shandong Agricultural University, Shandong, China. [7]Gothenburg Global Biodiversity Centre, Department of Biological and Environmental Sciences, University of Gothenburg, Göteborg, Sweden. [8]Department of Biology, University of Oxford, Oxford, UK. [9]Wuhan Botanical Garden, Chinese Academy of Sciences, Wuhan, China. [10]Institute of Molecular Plant Sciences, University of Edinburgh, Edinburgh, UK. [11]Department of Geography, University of Exeter, Exeter, UK. [12]Florida Museum of Natural History, University of Florida, Gainesville, FL, USA. [13]Biodiversity Institute, University of Florida, Gainesville, FL, USA. [14]Harvard University Herbaria, Department of Organismic and Evolutionary Biology, Harvard University, Cambridge, MA, USA. [15]Conservatoire et Jardin botaniques de Genève, Pregny-Chambésy, Switzerland. [16]School of Geosciences, Grant Institute, University of Edinburgh, Edinburgh, UK. [17]Department of Life Sciences and Systems Biology, University of Turin, Torino TO, Italy. [18]Department of Biodiversity and Conservation, Real Jardín Botánico (RJB), CSIC, Madrid, Spain. [19]Department of Biology, Case Western Reserve University, Cleveland, OH, USA. [20]Botanical Research Institute of Texas, Fort Worth, TX, USA. [21]Department of Anthropology, Penn State University, State College, PA, USA. [22]Herbarium and Center for Tree Science, The Morton Arboretum, Lisle, IL, USA. [23]School of Integrative Plant Science, Section of Plant Biology and the L.H. Bailey Hortorium, Cornell University, Ithaca, NY, USA. [24]Key Laboratory of Horticultural Plant Biology (Ministry of Education), College of Horticulture and Forestry Science, Huazhong Agricultural University, Wuhan, China. [25]International Center for Tropical Botany, Institute of Environment, Florida International University, Miami, FL, USA. [26]Commonwealth Scientific and Industrial Research Organisation (CSIRO), Australian National Insect Collection, Acton, ACT, Australia. [27]Department of Botany, University of Wisconsin-Madison, Madison, WI, USA. [28]Department of Computational Biology, Quartier Centre, University of Lausanne, Lausanne, Switzerland.

[29]Institute of Biology, University of Hohenheim, Stuttgart, Germany. [30]Biology Department, Westfield State University, Westfield, MA, USA. [31]Institute of Ecology and Evolution, University of Oregon, Eugene, OR, USA. [32]Department of Systematics, Biodiversity and Evolution of Plants (with Herbarium), University of Goettingen, Göttingen, Germany. [33]State Key Laboratory of Systematic and Evolutionary Botany, Institute of Botany, Chinese Academy of Sciences, Beijing, China. [34]Department of Plant and Microbial Biology, North Carolina State University, Raleigh, NC, USA. ✉e-mail: whuang@rbge.org.uk; Alex.Twyford@ed.ac.uk; phollingsworth@rbge.org.uk

