## [Transparent Peer Review file · Communications Biology]

DNA-based Identification of Plants and the Genomic Nature of Plant Species Differences

Corresponding Author: Dr Wu Huang

Version 0:

Reviewer comments:

Reviewer #1

(Remarks to the Author)

There is increasing demand for accurate species identification across different fields such as biodiversity research, forensics, and plant systematics. The study by Huang et al. addresses the potential of nuclear genomic data to offer better resolution for species identification as an alternative to traditional plant DNA barcoding. The latter often fails to distinguish closely related species and doesn't adequately address problems such as incomplete lineage sorting (ILS) or hybridization.

A total of 151 NGS-based studies employing multiple nuclear loci across a wide range of plant genera (including multiple individuals per species), using different sequencing techniques (target capture, genome skimming, transcriptome sequencing, and RAD/GBS), were explored. The proportion of monophyletic species, the frequency of species-specific single nucleotide polymorphisms (SSSNPs), and the minimum number of nuclear loci required for accurate species discrimination were calculated.

Their findings show that about 70% of the investigated plant species are monophyletic and possess distinct SSSNPs. The study also shows that approximately 200 loci or 3,000 SNPs are generally sufficient for high species resolution, while 1 to 10 well-chosen nuclear loci can be equally effective. Although whole-genome resequencing holds future promise, current methods still rely on targeted approaches. Huang et al. demonstrate that species-level identification usually does not require whole genomes or hundreds of nuclear markers.

The manuscript is well-structured and well-written.

Methods are adequately described, and all necessary details are provided in the manuscript or as supplementary material. Results are clearly and effectively presented.

The discussion also critically addresses issues that affect the current analysis, such as imperfect taxonomy, hybridization, ILS, and recent speciation, and emphasizes that e.g., the results presented here represent a broad approximate estimation. Future directions are discussed and recommendations for alternatives to the current DNA barcoding are provided.

Please correct line 250: *Taxus and Tsuga*, where "and" is currently in italics.

I am looking forward to see the manuscript published!

Reviewer #2

(Remarks to the Author)

This study utilizes diverse genomic datasets from 134 plant genera to differentiate species, a topic with significant potential impact as more public genomic data become available. However, there are some key questions that I believe should be addressed that could benefit the manuscript.

1. How does species differentiation using genomic data compare to traditional methods, such as standard DNA barcoding or locus-specific approaches? (1) could you compare phylogenetic trees for the same genus derived from genomic data versus those obtained through traditional methods? 2) could you evaluate and compare the phylogenetic support values (e.g., bootstrap support) derived from phylogenies reconstructed using genomic data versus those based on traditional datasets?

2. There is a difference in the datasets with respect to sequencing methods, as indicated in Table S1. Whole-genome sequencing (WGS) studies represent only 2.64% of the data, genome skimming studies account for 5.96%, while the dataset

is predominantly composed of reduced-representation approaches (54.96%). This imbalance could introduce biases into the results. Expanding the dataset to include a broader range of sequencing strategies, particularly whole-genome sequencing (WGS), could potentially affect the results shown in Figures 1 and 2. In this context, does whole-genome sequencing (WGS) provide any advantages? Publicly available datasets could be used to address this question.

3. Could you present an analysis showing how SSSNPs are differentiated by ploidy level?

Minor

1. Fig1 A. Revise the caption to indicate that the proportion refers to genera, not species.
2. Fig1 C. Could you include the number of monophyletic ratios represented in each boxplot within the figure?
3. What do the colors represent in Table S3?
4. In Figure 2: (1) Could you add the number of samples for each genus to the figure? (2) The GBS suffix for *Euphrasia* is not clearly visible. (3) Could you consider adding a tree for the genus to help group closely related genus within the figure?
5. Please add the number of species represented in each of the two boxplots in Figure S1?

Reviewer #3

(Remarks to the Author)
Comments to the Author

In the article entitled "DNA-based Identification of Plants and the Genomic Nature of Plant Species Differences" submitted for publication in *Communications Biology*, the authors conducted a large-scale reanalysis of sequencing data obtained through various methods in a total of 151 studies to investigate potential future strategies for plant DNA based identification. They assessed the density of species-specific SNP in plants, and found a median density of 193 SSSNPs per Mb and that a 895 had at least one SSSNP.

Using target capture dataset, they also found that 1-9 preselected loci could provide equivalent levels of species discrimination compared to 100s of nuclear loci in the initial datasets.

This well written study was carefully planned and well executed and will be an important foundation toward developing future approaches for DNA based plant species discrimination.

I have very little comments, and this manuscript could be published as it is in *Communications Biology*.

My only feedback would be about the method section:

Line 465: how did you identified these studies

Line 491: how did you estimated the density and abundance of the SSSNPs?

Line 551: which additional analyses were performed

Reviewer #4

(Remarks to the Author)

The manuscript "DNA-based Identification of Plants and the Genomic Nature of Plant Species Differences" presented by Huang and colleagues comprises a large study on DNA barcoding, monophyly, and species divergence in land plants. It was a pleasure to read such a well-written and comprehensive study with an elaborate discussion about their findings. Consequently, I only have one suggestion to improve this manuscript. In particular, I am curious about how the authors would define a DNA barcode region in the context of their work. As the manuscript emphasizes on species-specific SNPs, is the presence of species-specific SNPs a prerequisite for the selection of a suitable DNA barcode region? Specifying the concept of a DNA barcode region might help readers to correctly understand and interpret the outcome of this study and may facilitate the application of this work in future research.

Reviewer #5

(Remarks to the Author)

Version 1:

Reviewer comments:

Reviewer #2

(Remarks to the Author)

Reviewer #3

(Remarks to the Author)

*The authors revised their study following the feedback provided.
I recommend publication of the manuscript.*

Reviewer #4

(Remarks to the Author)

All my questions were sufficiently well answered. Thank you for the clarifications!

Open Access *This Peer Review File is licensed under a Creative Commons Attribution 4.0 International License, which permits use, sharing, adaptation, distribution and reproduction in any medium or format, as long as you give appropriate credit to the original author(s) and the source, provide a link to the Creative Commons license, and indicate if changes were made.*

Response to reviewers

Reviewer #comment	Reviewer's comments	Author's replies
1#1	There is increasing demand for accurate species identification across different fields such as biodiversity research, forensics, and plant systematics. The study by Huang et al. addresses the potential of nuclear genomic data to offer better resolution for species identification as an alternative to traditional plant DNA barcoding. The latter often fails to distinguish closely related species and doesn't adequately address problems such as incomplete lineage sorting (ILS) or hybridization..... The manuscript is well-structured and well-written. Methods are adequately described, and all necessary details are provided in the manuscript or as supplementary material. Results are clearly and effectively presented. The discussion also critically addresses issues that affect the current analysis, such as imperfect taxonomy, hybridization, ILS, and recent speciation, and emphasizes that e.g., the results presented here represent a broad approximate estimation. Future directions are discussed and recommendations for alternatives to the current DNA barcoding are provided. Please correct line 250: Taxus and Tsuga, where "and" is currently in italics. I am looking forward to see the manuscript published!	Response: Thank you for your comments we appreciate the reviewers' thoughts and positive commentary on the manuscript. In terms of the specific formatting issue, we have un-italicised "and" in line 250 (currently line 251).
2#1	This study utilizes diverse genomic datasets from 134 plant genera to differentiate species, a topic with significant potential impact as more public genomic data become available. However, there are some key questions	We agree with the reviewer in the value of comparing our results from multiple nuclear loci to standard DNA barcoding approaches. Direct comparisons are difficult because there are relatively few studies which have directly compared between standard barcode markers and multiple nuclear loci, and where the sampling includes multiple individuals from multiple congeneric species.

	that I believe should be addressed that could benefit the manuscript. 1. How does species differentiation using genomic data compare to traditional methods, such as standard DNA barcoding or locus-specific approaches? 2. could you compare phylogenetic trees for the same genus derived from genomic data versus those obtained through traditional methods? ----- 2) could you evaluate and compare the phylogenetic support values (e.g., bootstrap support) derived from phylogenies reconstructed using genomic data versus those based on traditional datasets?	However suitable comparisons are possible in a few specific case studies. WGS sequencing produces data for ITS and the plastome alongside nuclear regions in the same dataset. Some hyb-seq studies also recovered off-target sequences from which the plastome can be recovered. We recovered 12 suitable datasets and in these the nuclear data sets showed an increase in species discrimination compared to standard barcodes in 83% of cases (10/12 studies, with no difference in absolute discrimination in the remaining two cases). Table S12 has been added to illustrate this point and is referred to at line 418-421. ----- Our focus is purely on the whether a species resolves as monophyletic or not. In recently diverged species, there may be a very limited number of characters involved, and this may be associated with low bootstrap support, but this would not negate the species level monophyly per se. Thus, rather than include bootstrap thresholds, we focus purely on non-monophyly versus monophyly. This enables comparison across the wide range of datasets we have included in our study and different methods of analysis used by the different original study authors (who have used different tree-building algorithms and different approaches for reporting topological support which would also confound bootstrap support comparisons).
2#2	There is a difference in the datasets with respect to sequencing methods, as indicated in Table S1. Whole-genome sequencing (WGS) studies represent only 2.64% of the data, genome skimming studies account for 5.96%, while the dataset is predominantly composed of reduced-representation approaches (54.96%). This imbalance could introduce biases into the results. Expanding the dataset to include a broader range of sequencing strategies, particularly whole-genome sequencing (WGS), could potentially affect the results shown in Figures 1 and 2. In this context, does whole-genome sequencing (WGS) provide any advantages? Publicly available datasets could be used to address this question.	Our analysis reflected the availability of datasets that included multiple individuals of multiple congeneric species (and were focused on wild (not domesticated) species). There remain only a small number of subsequent whole genome sequencing studies that meet our sampling requirement. The over-riding point for Fig. 1 is a quantification of the scale of plant species non-monophyly. We do not expect to see a materially different outcome from more whole genome sequencing studies. Our in silico sub-sampling analysis shows an asymptote in recovery of species monophyly at around 1500-3000 SNPs and represents an empirical test of this issue. Further supporting this – in our more recent follow-up work where we have produced reference genomes, and whole genome resequencing data from test genera, and sub-sampled the data – we again do not see material differences in resolving power between the WGS data and subsets of it (e.g. in silico extraction of BUSCO or 353 loci).

		For Fig. 2 – we agree that the results on the density of Species-specific SNPs are more susceptible to the influence of different sequencing approaches – but note that a considerable amount more data will be required to materially refine the density estimates including accommodating the additional sources of variance including different taxonomic groups, ages of species divergence, life history forms, and variation in the density of species-sampled-per-genus among studies. Simply adding the 4-5 recently generated WGS datasets we have found (which have a strong focus on domestication which might adding to another level of bias) will not allow a robust test of differences compared to other sequencing strategies. We also note our existing data has good representation of target capture datasets (focusing on conserved/coding regions) and RAD/GBS data (mostly representing non-coding and genome wide variation), thus encompassing broad representation of plant genomes. In our original submission we made clear the caveats around the density of SSSNPs, and reflecting the reviewer’s comment have strengthened this further. Thus, our preference is to adjust the manuscript with a further strengthening of the caveat, rather than delaying its publication by re-running a time-consuming set of analyses on a handful of newly available studies which ultimately will only make an incremental addition to the current findings, and will not address the substantive point that has been raised the reviewer.
2#3	Could you present an analysis showing how SSSNPs are differentiated by ploidy level?	This is an interesting question but out of scope of the manuscript and was not an aim of our study, and our sample set is not designed with this in mind. Even reliably assessing ploidy level from this diverse set of genera is challenging. In our experience of curating our own data sets, the frequent discrepancies between published chromosome counts and genome size at an individual specimen level, along with the subsequent step of assigning specimens to a given ploidy level gives us strong reservations about adding in this level of uncertainty across multiple genera at a manuscript revision stage. As an addendum - we do cover the likely influence of polyploidy as one of the causes of the taxa not showing monophyly despite using a large number of loci in line 332-336 and line 445-447.
2#4	Fig1 A. Revise the caption to indicate that the proportion refers to genera, not species.	Line 197: We added ‘within a genus’ to the caption of Fig.1 A.

2#5	Fig1 C. Could you include the number of monophyletic ratios represented in each boxplot within the figure?	Thank you for pointing this out. We added the number of data points (n) of each boxplots on the Fig1 B and C. (Please find below in Revised figures and in the manuscript.)
2#6	What do the colors represent in Table S3?	Sorry for the confusion of the colouring highlights in Table S3. They were for our own use in tracking multiple genera in the same family. We have removed the highlights.
2#7	In Figure 2: (1) Could you add the number of samples for each genus to the figure? (2) The GBS suffix for Euphrasia is not clearly visible.	We have added the number of multiple-sampled species (n) of each genus to Fig. 2 and add the GBS suffix to the associated genus Euphrasia .
2#8	Could you consider adding a tree for the genus to help group closely related genus within the figure?	Thank you for the suggestion. We have added a tree based on the Tree of Life family tree version 3.0 for our groups and reordered the genera in Fig. 2 and added a note to this in the Materials and Methods (Line 516-517).
3#1	This well written study was carefully planned and well executed and will be an important foundation toward developing future approaches for DNA based plant species discrimination. I have very little comments, and this manuscript could be published as it is in Communications Biology. My only feedback would be about the method section: Line 465: how did you identified these studies	In the main text following line 465 (currently 466), we have provided a link to our detailed protocol outlining the full inclusion criteria and study identification process: http://dx.doi.org/10.17504/protocols.io.kxygx3z9og8j/v1 . This protocol describes the search strategy, screening steps, and criteria used to identify eligible studies. We have clarified this in the manuscript to ensure the process is more immediately transparent to readers. In a nutshell, we used ambiguous matching patterns to search in the Web of Science and University of Edinburgh literature search engines for journal publications for studies that sequenced multiple loci from the nuclear genome and which sampled multiple individuals of multiple congeneric species. Details are described in section 1.1 in the protocol. We then filtered the publications by criteria described in section 1.2 in the protocol.
3#2	Line 491: how did you estimated the density and abundance of the SSSNPs?	After identifying SNPs from the alignments, we used the script “extract_ancestral_informative_SNPs_vcf.py” to extract species-specific-SNPs which are fixed in all individuals from one species and distinct from all other ingroup species. The analytical scripts used can be found at https://github.com/Hazelhuangup/NucBarcode . Further details on the methods used to estimate the density and abundance of SSSNPs are provided in our analysis protocol, available at: http://dx.doi.org/10.17504/protocols.io.5qpvo33rv4o/v1 . Specifically in section 5 “The extraction of taxonomically informative Loci (Species-specific SNPs)”. It outlines the analytical steps and criteria applied in the estimation process.
3#3	Line 551: which additional analyses were performed	We have revised the manuscript to clarify these additional analyses. Specifically, for each dataset, we assessed which individual genes recovered the maximum

		proportion of species as monophyletic (measured by species monophyly) and which minimal combinations of individual genes could recover the same number of monophyletic species as the complete dataset. This is added to the manuscript on line 538-542 along with a link to a more detailed analysis protocol.
4#1	The manuscript “DNA-based Identification of Plants and the Genomic Nature of Plant Species Differences” presented by Huang and colleagues comprises a large study on DNA barcoding, monophyly, and species divergence in land plants. It was a pleasure to read such a well-written and comprehensive study with an elaborate discussion about their findings. Consequently, I only have one suggestion to improve this manuscript. In particular, I am curious about how the authors would define a DNA barcode region in the context of their work. As the manuscript emphasizes on species-specific SNPs, is the presence of species-specific SNPs a prerequisite for the selection of a suitable DNA barcode region? Specifying the concept of a DNA barcode region might help readers to correctly understand and interpret the outcome of this study and may facilitate the application of this work in future research.	The well-established definition of a DNA barcode that we use is short standardised region(s) of DNA sequence that can be used to reliably identify species. The ideal DNA barcode should balance universality, recoverability, and species-level resolution (e.g. CBOL Plant Working Group, 2009). However, we also recognise there is an increasing interest in extending the concept of DNA barcoding to include wider genomic approaches, see Coissac et. al. (2016), Hollingsworth et. al. (2016), Here we make no attempt to define what a nuclear barcode would look like and how it should be defined (although in our view, were that to be done, we indeed would not restrict any such definition to requiring ubiquitous presence of species-specific SNPs). Instead, we are researching and quantifying the nature of differences between plant species to provide information that would guide the future downstream definition and refinement of new genomic barcodes. An approximation of the density of species-specific SNPs provides insights into how complicated this task will be - the greater the density of species-specific SNPs the easier it will be to develop new barcoding methods involving small numbers of loci from the nuclear genome. Reflecting the reviewers comment we added some phrases to the discussion to try and bring out the point that the downstream definition of a genomic barcode is for the future and not for this manuscript. CBOL, Plant Working Group, A DNA barcode for land plants. Proceedings of the National Academy of Sciences 106, 12794-12797 (2009). E. Coissac, P. M. Hollingsworth, S. Lavergne, P. Taberlet, From barcodes to genomes: extending the concept of DNA barcoding. Molecular Ecology 25, 1423-1428 (2016). P. M. Hollingsworth, D. Z. Li, M. van der Bank, A. D. Twyford, Telling plant species apart with DNA: from barcodes to genomes. Philosophical Transactions of the Royal Society B: Biological Sciences 371 (2016).

Revised figures:

Fig. 1. Quantification of the proportion of plant species within a genus that resolve as monophyletic based on nuclear genomic data for 151 studies. (A) Proportion of species resolving as monophyletic in each study (B) Comparison of plant groups with herbaceous and woody life forms (C)

Comparison of different sequencing methods. Boxplots show the median, lower, and upper quartiles, with whiskers extending to 1.5 times the interquartile range; the dots in (C) are the monophyletic ratio for each study.

Changes:

1. We have added 'within a genus' to the caption of Fig1 A.
2. We have added the number of data points (n) of each boxplots on the Fig1 B and C.

Fig. 2. Distribution of the density of species-specific SNPs (SSSNPs) for 27 datasets from 26 genera (left panel, log-transformed), with the phylogenetic relationships among study taxa (middle panel), and the proportion of species that have more than one SSSNP (right panel). The right panel is the proportion of species in each genus that have more than one SSSNP (orange bars) and the proportion of species that resolve as monophyletic in that genus (blue bars).

Changes:

1. We have added the number of multiple-sampled species (n) of each genus to Fig. 2 and add the GBS suffix to the associated genus *Euphrasia*.
2. We have added a tree based on the Tree of Life family tree version 3.0 for our groups and reordered the genera in Fig. 2 and added a note to this in the Materials and Methods (Line 516-517).

Response to reviewers

Reviewer #comment	Reviewer's comments	Author's replies
1#1	There is increasing demand for accurate species identification across different fields such as biodiversity research, forensics, and plant systematics. The study by Huang et al. addresses the potential of nuclear genomic data to offer better resolution for species identification as an alternative to traditional plant DNA barcoding. The latter often fails to distinguish closely related species and doesn't adequately address problems such as incomplete lineage sorting (ILS) or hybridization..... The manuscript is well-structured and well-written. Methods are adequately described, and all necessary details are provided in the manuscript or as supplementary material. Results are clearly and effectively presented. The discussion also critically addresses issues that affect the current analysis, such as imperfect taxonomy, hybridization, ILS, and recent speciation, and emphasizes that e.g., the results presented here represent a broad approximate estimation. Future directions are discussed and recommendations for alternatives to the current DNA barcoding are provided. Please correct line 250: Taxus and Tsuga, where "and" is currently in italics. I am looking forward to see the manuscript published!	Response: Thank you for your comments we appreciate the reviewers' thoughts and positive commentary on the manuscript. In terms of the specific formatting issue, we have un-italicised "and" in line 250 (currently line 251).
2#1	This study utilizes diverse genomic datasets from 134 plant genera to differentiate species, a topic with significant potential impact as more public genomic data become available. However, there are some key questions	We agree with the reviewer in the value of comparing our results from multiple nuclear loci to standard DNA barcoding approaches. Direct comparisons are difficult because there are relatively few studies which have directly compared between standard barcode markers and multiple nuclear loci, and where the sampling includes multiple individuals from multiple congeneric species.

	that I believe should be addressed that could benefit the manuscript. 1. How does species differentiation using genomic data compare to traditional methods, such as standard DNA barcoding or locus-specific approaches? 2. could you compare phylogenetic trees for the same genus derived from genomic data versus those obtained through traditional methods? ----- 2) could you evaluate and compare the phylogenetic support values (e.g., bootstrap support) derived from phylogenies reconstructed using genomic data versus those based on traditional datasets?	However suitable comparisons are possible in a few specific case studies. WGS sequencing produces data for ITS and the plastome alongside nuclear regions in the same dataset. Some hyb-seq studies also recovered off-target sequences from which the plastome can be recovered. We recovered 12 suitable datasets and in these the nuclear data sets showed an increase in species discrimination compared to standard barcodes in 83% of cases (10/12 studies, with no difference in absolute discrimination in the remaining two cases). Table S12 has been added to illustrate this point and is referred to at line 418-421. ----- Our focus is purely on the whether a species resolves as monophyletic or not. In recently diverged species, there may be a very limited number of characters involved, and this may be associated with low bootstrap support, but this would not negate the species level monophyly per se. Thus, rather than include bootstrap thresholds, we focus purely on non-monophyly versus monophyly. This enables comparison across the wide range of datasets we have included in our study and different methods of analysis used by the different original study authors (who have used different tree-building algorithms and different approaches for reporting topological support which would also confound bootstrap support comparisons).
2#2	There is a difference in the datasets with respect to sequencing methods, as indicated in Table S1. Whole-genome sequencing (WGS) studies represent only 2.64% of the data, genome skimming studies account for 5.96%, while the dataset is predominantly composed of reduced-representation approaches (54.96%). This imbalance could introduce biases into the results. Expanding the dataset to include a broader range of sequencing strategies, particularly whole-genome sequencing (WGS), could potentially affect the results shown in Figures 1 and 2. In this context, does whole-genome sequencing (WGS) provide any advantages? Publicly available datasets could be used to address this question.	Our analysis reflected the availability of datasets that included multiple individuals of multiple congeneric species (and were focused on wild (not domesticated) species). There remain only a small number of subsequent whole genome sequencing studies that meet our sampling requirement. The over-riding point for Fig. 1 is a quantification of the scale of plant species non-monophyly. We do not expect to see a materially different outcome from more whole genome sequencing studies. Our in silico sub-sampling analysis shows an asymptote in recovery of species monophyly at around 1500-3000 SNPs and represents an empirical test of this issue. Further supporting this – in our more recent follow-up work where we have produced reference genomes, and whole genome resequencing data from test genera, and sub-sampled the data – we again do not see material differences in resolving power between the WGS data and subsets of it (e.g. in silico extraction of BUSCO or 353 loci).

		For Fig. 2 – we agree that the results on the density of Species-specific SNPs are more susceptible to the influence of different sequencing approaches – but note that a considerable amount more data will be required to materially refine the density estimates including accommodating the additional sources of variance including different taxonomic groups, ages of species divergence, life history forms, and variation in the density of species-sampled-per-genus among studies. Simply adding the 4-5 recently generated WGS datasets we have found (which have a strong focus on domestication which might adding to another level of bias) will not allow a robust test of differences compared to other sequencing strategies. We also note our existing data has good representation of target capture datasets (focusing on conserved/coding regions) and RAD/GBS data (mostly representing non-coding and genome wide variation), thus encompassing broad representation of plant genomes. In our original submission we made clear the caveats around the density of SSSNPs, and reflecting the reviewer’s comment have strengthened this further. Thus, our preference is to adjust the manuscript with a further strengthening of the caveat, rather than delaying its publication by re-running a time-consuming set of analyses on a handful of newly available studies which ultimately will only make an incremental addition to the current findings, and will not address the substantive point that has been raised the reviewer.
2#3	Could you present an analysis showing how SSSNPs are differentiated by ploidy level?	This is an interesting question but out of scope of the manuscript and was not an aim of our study, and our sample set is not designed with this in mind. Even reliably assessing ploidy level from this diverse set of genera is challenging. In our experience of curating our own data sets, the frequent discrepancies between published chromosome counts and genome size at an individual specimen level, along with the subsequent step of assigning specimens to a given ploidy level gives us strong reservations about adding in this level of uncertainty across multiple genera at a manuscript revision stage. As an addendum - we do cover the likely influence of polyploidy as one of the causes of the taxa not showing monophyly despite using a large number of loci in line 332-336 and line 445-447.
2#4	Fig1 A. Revise the caption to indicate that the proportion refers to genera, not species.	Line 197: We added ‘within a genus’ to the caption of Fig.1 A.

2#5	Fig1 C. Could you include the number of monophyletic ratios represented in each boxplot within the figure?	Thank you for pointing this out. We added the number of data points (n) of each boxplots on the Fig1 B and C. (Please find below in Revised figures and in the manuscript.)
2#6	What do the colors represent in Table S3?	Sorry for the confusion of the colouring highlights in Table S3. They were for our own use in tracking multiple genera in the same family. We have removed the highlights.
2#7	In Figure 2: (1) Could you add the number of samples for each genus to the figure? (2) The GBS suffix for Euphrasia is not clearly visible.	We have added the number of multiple-sampled species (n) of each genus to Fig. 2 and add the GBS suffix to the associated genus Euphrasia .
2#8	Could you consider adding a tree for the genus to help group closely related genus within the figure?	Thank you for the suggestion. We have added a tree based on the Tree of Life family tree version 3.0 for our groups and reordered the genera in Fig. 2 and added a note to this in the Materials and Methods (Line 516-517).
3#1	This well written study was carefully planned and well executed and will be an important foundation toward developing future approaches for DNA based plant species discrimination. I have very little comments, and this manuscript could be published as it is in Communications Biology. My only feedback would be about the method section: Line 465: how did you identified these studies	In the main text following line 465 (currently 466), we have provided a link to our detailed protocol outlining the full inclusion criteria and study identification process: http://dx.doi.org/10.17504/protocols.io.kxygx3z9og8j/v1 . This protocol describes the search strategy, screening steps, and criteria used to identify eligible studies. We have clarified this in the manuscript to ensure the process is more immediately transparent to readers. In a nutshell, we used ambiguous matching patterns to search in the Web of Science and University of Edinburgh literature search engines for journal publications for studies that sequenced multiple loci from the nuclear genome and which sampled multiple individuals of multiple congeneric species. Details are described in section 1.1 in the protocol. We then filtered the publications by criteria described in section 1.2 in the protocol.
3#2	Line 491: how did you estimated the density and abundance of the SSSNPs?	After identifying SNPs from the alignments, we used the script “extract_ancestral_informative_SNPs_vcf.py” to extract species-specific-SNPs which are fixed in all individuals from one species and distinct from all other ingroup species. The analytical scripts used can be found at https://github.com/Hazelhuangup/NucBarcode . Further details on the methods used to estimate the density and abundance of SSSNPs are provided in our analysis protocol, available at: http://dx.doi.org/10.17504/protocols.io.5qpvo33rv4o/v1 . Specifically in section 5 “The extraction of taxonomically informative Loci (Species-specific SNPs)”. It outlines the analytical steps and criteria applied in the estimation process.
3#3	Line 551: which additional analyses were performed	We have revised the manuscript to clarify these additional analyses. Specifically, for each dataset, we assessed which individual genes recovered the maximum

		proportion of species as monophyletic (measured by species monophyly) and which minimal combinations of individual genes could recover the same number of monophyletic species as the complete dataset. This is added to the manuscript on line 538-542 along with a link to a more detailed analysis protocol.
4#1	The manuscript “DNA-based Identification of Plants and the Genomic Nature of Plant Species Differences” presented by Huang and colleagues comprises a large study on DNA barcoding, monophyly, and species divergence in land plants. It was a pleasure to read such a well-written and comprehensive study with an elaborate discussion about their findings. Consequently, I only have one suggestion to improve this manuscript. In particular, I am curious about how the authors would define a DNA barcode region in the context of their work. As the manuscript emphasizes on species-specific SNPs, is the presence of species-specific SNPs a prerequisite for the selection of a suitable DNA barcode region? Specifying the concept of a DNA barcode region might help readers to correctly understand and interpret the outcome of this study and may facilitate the application of this work in future research.	The well-established definition of a DNA barcode that we use is short standardised region(s) of DNA sequence that can be used to reliably identify species. The ideal DNA barcode should balance universality, recoverability, and species-level resolution (e.g. CBOL Plant Working Group, 2009). However, we also recognise there is an increasing interest in extending the concept of DNA barcoding to include wider genomic approaches, see Coissac et. al. (2016), Hollingsworth et. al. (2016), Here we make no attempt to define what a nuclear barcode would look like and how it should be defined (although in our view, were that to be done, we indeed would not restrict any such definition to requiring ubiquitous presence of species-specific SNPs). Instead, we are researching and quantifying the nature of differences between plant species to provide information that would guide the future downstream definition and refinement of new genomic barcodes. An approximation of the density of species-specific SNPs provides insights into how complicated this task will be - the greater the density of species-specific SNPs the easier it will be to develop new barcoding methods involving small numbers of loci from the nuclear genome. Reflecting the reviewers comment we added some phrases to the discussion to try and bring out the point that the downstream definition of a genomic barcode is for the future and not for this manuscript. CBOL, Plant Working Group, A DNA barcode for land plants. Proceedings of the National Academy of Sciences 106, 12794-12797 (2009). E. Coissac, P. M. Hollingsworth, S. Lavergne, P. Taberlet, From barcodes to genomes: extending the concept of DNA barcoding. Molecular Ecology 25, 1423-1428 (2016). P. M. Hollingsworth, D. Z. Li, M. van der Bank, A. D. Twyford, Telling plant species apart with DNA: from barcodes to genomes. Philosophical Transactions of the Royal Society B: Biological Sciences 371 (2016).

Revised figures:

Fig. 1. Quantification of the proportion of plant species within a genus that resolve as monophyletic based on nuclear genomic data for 151 studies. (A) Proportion of species resolving as monophyletic in each study (B) Comparison of plant groups with herbaceous and woody life forms (C)

Comparison of different sequencing methods. Boxplots show the median, lower, and upper quartiles, with whiskers extending to 1.5 times the interquartile range; the dots in (C) are the monophyletic ratio for each study.

Changes:

1. We have added 'within a genus' to the caption of Fig1 A.
2. We have added the number of data points (n) of each boxplots on the Fig1 B and C.

Fig. 2. Distribution of the density of species-specific SNPs (SSSNPs) for 27 datasets from 26 genera (left panel, log-transformed), with the phylogenetic relationships among study taxa (middle panel), and the proportion of species that have more than one SSSNP (right panel). The right panel is the proportion of species in each genus that have more than one SSSNP (orange bars) and the proportion of species that resolve as monophyletic in that genus (blue bars).

Changes:

1. We have added the number of multiple-sampled species (n) of each genus to Fig. 2 and add the GBS suffix to the associated genus *Euphrasia*.
 2. We have added a tree based on the Tree of Life family tree version 3.0 for our groups and reordered the genera in Fig. 2 and added a note to this in the Materials and Methods (Line 516-517).
-
1. a tree based on the Tree of Life family tree version 3.0 for our groups and reordered the genera